# Usefulness of Scissors with a Power-Support Mechanism to Assist Thumb Movement: An Observational Study

**Kohei Koizumi** [1], **Kumiko Sasao** [1], **Yuji Koike** [1], **Akihisa Okino** [2], **Kazuhisa Takeda** [2] and **Toyohiro Hamaguchi** [1,*]

1   Department of Rehabilitation, Graduate School of Health Sciences, Saitama Prefectural University, Koshigaya 3438540, Saitama, Japan; koizumi-kohei@spu.ac.jp (K.K.); sasao-kumiko@spu.ac.jp (K.S.); koike-yuji@spu.ac.jp (Y.K.)
2   Okino Robotics Industries, Ltd., Kamikawa 3670241, Saitama, Japan; okino@okino-robotics.co.jp (A.O.); takeda@okino-robotics.co.jp (K.T.)
*   Correspondence: hamaguchi-toyohiro@spu.ac.jp; Tel.: +81-48-973-4125

**Abstract:** Long-term repetitive movements, such as opening and closing scissors, increase strain on muscles and joints. Amplitude probability distribution function (APDF) analysis of surface electromyogram (sEMG) data was used to quantify the burden of muscle activity. We aimed to test the hypothesis that scissors with a power-support device assist repetitive thumb movements to reduce potential myoelectric activity. Twenty female university students who met the eligibility criteria performed a cutting experiment, with and without power-support device scissors. The primary outcome was a change in muscle load due to sEMG data that were analyzed using APDF, and the secondary outcomes investigated the occurrence of muscle fatigue and pain. The adductor pollicis muscle showed a significant decrease in muscle activity with power assistance. In addition, it was also found that fatigue and pain of the thumb and on the radial side of the forearm were significantly lower under the power-assisted conditions. The results of this study suggest that the assistive action of scissors with a power-support device compensate for muscle load on the thenar eminence. This may be used as a reference value to prevent the occurrence of hand disorders for hairdressers.

**Keywords:** scissors; adductor pollicis muscle; surface electromyogram; amplitude probability distribution function; pain; fatigue

## 1. Introduction

During repeated occupational exertions, such as when a hairdresser is cutting with scissors, a burden is placed upon the bones, joints, muscles, tendons, ligaments, and peripheral nerves, which may cause musculoskeletal disorders, including tendonitis of the hand [1]. Upon analysis of hairdressers' time spent performing duties, cutting with scissors predominates, taking up an average of 29% of the time [2]. The excessive repetitive movements involved in manipulating scissors for long periods are often associated with carpal tunnel syndrome (CTS) and De Quervain's tenosynovitis [3,4]. Disorders due to overuse of the hands are common in occupations, such as hairdressers, that use scissors continuously for long periods [5,6], and hand disorders are a cause of hairdressers leaving the profession [7].

Carpal tunnel syndrome is caused by repetitive finger movements, such as when opening and closing scissors. Carpal tunnel pressure is lowest in the intermediate position and highest in the dorsiflexion position of the wrist joint [8]. Therefore, it is recommended to maintain the wrist joint in the neutral position to reduce pressure in the carpal tunnel when CTS occurs. The wrist joint and forearm maintain an almost neutral position during the opening and closing movements when cutting paper [9], but hairdressers use their forearm in the pronated position and wrist joint in the flexed and dorsiflexed position. Therefore, the pressure in the carpal tunnel is high, and the incidence of muscle malaise and pain is high [4].

Small scissors are mainly held in the dominant hand, and cut an object by the opening and closing movements of the thumb and index, middle, and ring fingers [9,10]. A hairdresser bends the thumb to close the scissors and uses the weight of the handle to open the blades; this movement is mainly performed by the thenar eminence and flexor pollicis longus. Especially, when the lower blade of the barber scissors is used to cut, the thumb is fixed into the handle ring and the adductor pollicis performs the motion of the thumb. Fixation of the thumb into the handle ring adjusts the fit to the ring diameter by slightly acting on the flexor pollicis longus muscle. The opening and closing movement of the scissors is mainly performed by the adductor pollicis muscle [11]. The shape of scissors has not changed since ancient times [12], because there is no substitute for the associated movement-cutting efficiency. There is a report that the shape of the scissors was changed based on ergonomics in an attempt to reduce the burden on the hands of cosmetologists [13]. Although the pain in the wrist joint was alleviated, the load on the forearm and fingers due to repetitive exercise was not verified, and the problem remained. Since 2017, we have developed scissors with a power-support device to assist with extracorporeal power and flexion of the thumb, to supplement the contractive force of the thenar muscle during repetitive exercise, and reduce exercise load. This reduction in load may help prevent musculoskeletal disorders, but it remains unclear to what extent the scissors can assist in increasing the force of the intrinsic muscles upon cutting.

Although previous research quantifies musical instrument performance, typing, and smartphone operation by combining various measurement conditions and muscle activity [14,15], there is no report that quantitatively measures the burden on the wrist joint and fingers during the opening and closing movements of scissors. To optimize the operation of scissors to prevent hairdressers work-related obstacles, it is necessary to understand muscle activity and muscle mobilization patterns during a particular task. To determine this, muscle activity using a surface electromyogram (sEMG) is utilized, and analyzed using amplitude probability distribution function (APDF). This can then be used as an effective reference value for the prevention of disabilities for hairdressers.

In this study, we used APDF analysis to determine whether scissors with a power-support device can improve the repetitive movement of the thumb to reduce the amount of myoelectric potential activity. Our secondary aim was to examine the effect of these scissors on muscle fatigue and pain. Our hypothesis was that scissors with a power-support device can effectively assist thumb movements and reduce the burden on the musculoskeletal system. A reference value can then be determined, to measure and prevent tendonitis and CTS, even with the long-term use of barber scissors. Alternatively, the kinematic findings related to the use of scissors to prevent the occurrence of disorders can be obtained.

## 2. Materials and Methods

### 2.1. Study Design

This cross-sectional study compared the difference between power-supported scissors and non-power-supported scissors, using the muscle activity of the adductor pollicis muscle (ADP) and extensor carpi radialis muscle (ECR) as indicators. The ADP muscle was used as the analysis target muscle because it is the main muscle involved in the opening and closing movement of scissors. The reason for measuring the ECR was to confirm whether the dorsiflexion position of the wrist joint, which affects the movement of the thumb, is maintained. All participants provided informed consent after the purpose of the study was explained in detail.

### 2.2. Participants and Recruitment

The inclusion criteria were female university students over 18 years of age, who were right-handed, and without movement and sensory disorders in the upper limbs. Only women were included because of the high number of women with CTS [6,16]. The exclusion criterion was those who were suspected of having CTS in the pre-experimental examination. The participant's dominant hand was determined to be right, with a score

of 80 or higher using The Edinburgh Handedness Inventory (EHI) [17,18]. Carpal tunnel syndrome symptoms before the experiment were determined to be asymptomatic with a score of 0 using The Japanese Society for Surgery of the Hand version of the Carpal Tunnel Syndrome Instrument (CTSI-JSSH) [19].

The number of participants required for the analysis of this study was 20, based on the report of electromyogram (EMG) analysis [20], assuming an error of 100 mV, and a standard deviation of 200 mV, with the minimum value calculated to be 18 with a reliability of 95%.

We recruited university students from Saitama Prefectural University, via email, between April 2020 and March 2021, to verify the effectiveness of the power-support scissors. The study procedure was explained to subjects who met the selection criteria and consented to participate. Students were informed of their right to withdraw from the study at any time, without prejudice, and key characteristics and clinical data were obtained at the time of registration. Participants performed the EHI evaluation for handedness and CTSI-JSSH, after providing information on height, weight, and body mass index (BMI). Prior to the experiment, the subjects were informed of how to use the scissors and precautions. After that, the handle ring was adjusted according to the circumference of the thumb, and the scissors opening and closing movement was practiced. When a gap was found between the circumference of the thumb and the handle ring, it was adjusted with a silicone finger hole ring.

### 2.3. Experimental Setup

For the upper limb position at the time of measurement, the subject was seated in a chair, with the right elbow placed on a table. The scissors were held, with the shoulder joint at 45 degrees of abduction, the elbow joint at 90 degrees of flexion, the forearm pronated, and the wrist at 30 degrees of dorsiflexion. During the cutting task, we presented scissors with the blades opened at 30 degrees, so that reproducibility of the angles was maintained. Participants were asked to repeatedly open and close the scissors (empty cut) to the rhythm of an electronic metronome (tempo: 2.0 Hz) output from a computer.

A preliminary experiment was conducted, with a hairdresser performing 100 consecutive open/close movements, and marginal subjective muscle fatigue of the fingers recorded. A video was taken of the hairdresser cutting hair (cutting point 30–50 mm from the cutting edge), and the opening and closing frequency of the scissors was measured as 2.0–3.0 Hz. This aimed to be a preliminary experiment using the newly developed equipment to verify the effect of the power-support scissors. The task repetition frequency was subsequently set to 2.0 Hz, and the scissors opening/closing operation was performed 200 times, with and without assistance, for a total of 400 operations. Furthermore, the exercise time interval was set to 180 s, with reference to past experiments [21], so that fatigue would not influence the experimental data.

### 2.4. Procedure

The protocol for measuring the open/close motion was as follows. First, the maximum isometric contraction of the ADP and ECR was simultaneously measured using a pinch meter (MT-140, Sakai Medical, Tokyo), before the open/close motion was measured for 8 s. The electromyographic data at the time of maximum contraction of each muscle was recorded as the maximum voluntary isometric contraction (MVIC) force. Next, after 180 s of rest, and confirmation that there was no pain or subjective fatigue of the ADP and ECR, the scissors opening/closing movement was repeated 200 times (task execution time: 100 s) at a rate of 2 times per second (2.0 Hz) (Figure 1). The exercise was performed with and without assistance, in a randomized order, with a 180-s rest between conditions, and pain and subjective fatigue of the ADP and ECR were measured by visual analog scale (VAS). A higher score denoted more pain or fatigue.

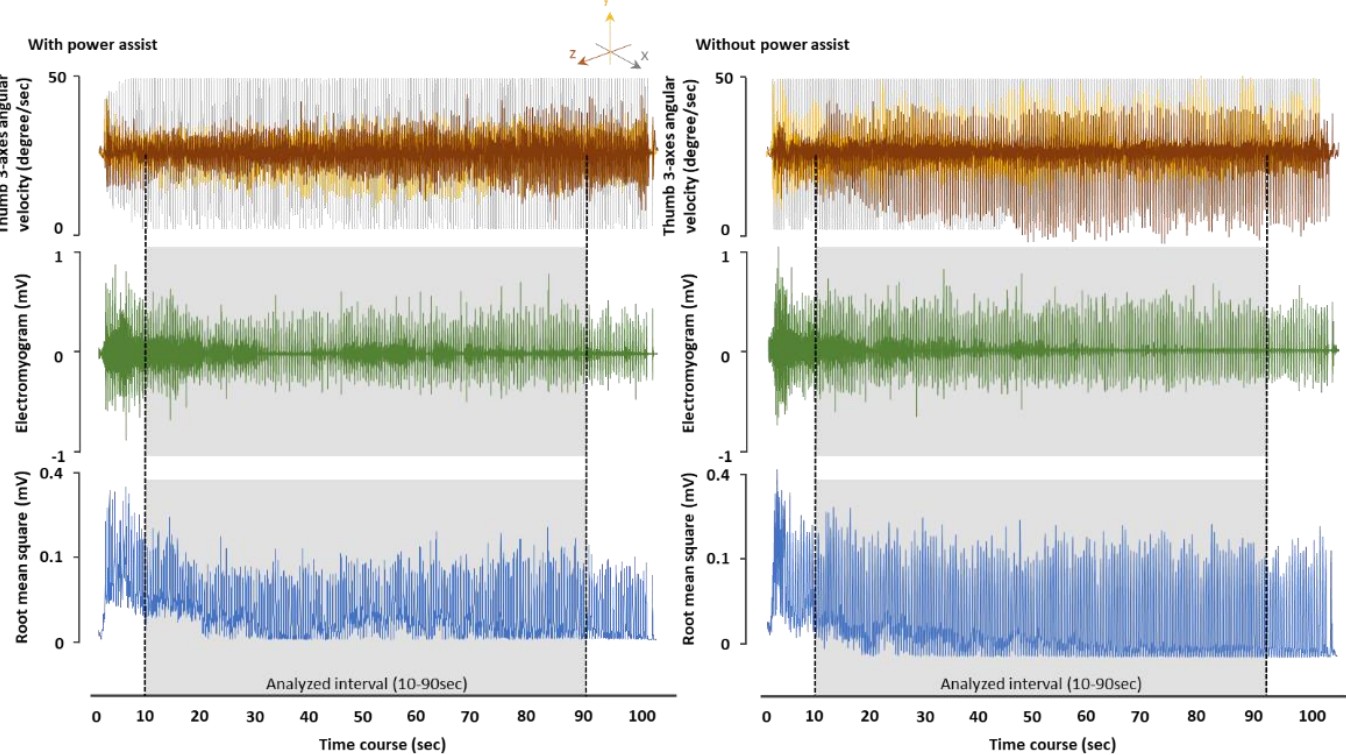

**Figure 1.** Thumb angular velocity (3 axes) during scissors opening and closing. The raw−surface electromyogram (sEMG) and root mean square (RMS) envelope derived from the adductor pollicis muscle (ADP) are shown in a single sample. Left panel, when power assistance is applied; right panel, no power assistance. The thumb 3−axes angular velocity is shown: gray line, x−axis; yellow, y−axis; brown, z−axis. Data was acquired for 100 s, of which 10–90 s were analyzed.

### 2.5. Power-Support Scissors Mechanism

The barber scissors used in this study were RAY COSMOS (2019, Hikari, Tokyo, Japan), as these have a standard shape, with a 93 mm stainless-steel blade, a total length of 176 mm, and a blade thickness of 3 mm. For power support, the same scissors were mounted with a power-assist mechanism. The scissors used under each condition did not have a significant weight difference (without assistance, 66 g; with assistance, 71 g). The power-assist mechanism provides assistance by connecting the ring of the lower blade and the shaft support of both blades with a steel wire (Patent Application No. 2020-054533). The wire is stretched when the blade is opened, and the restoring force assists the blade when it is closed. The reference value of the assistance obtained from the restoring force of the wire was recorded as the tension of the wire measured with a digital push-pull gauge (RX-10, Aikoh Engineering, Osaka) when opening/closing the scissors (open angles: 30 and 50 degrees). The resulting force for the open angles was 0.81 N at 30 degrees and 1.15 N at 50 degrees. The torque score calculated from the assist force of the wire and the point of action (Distance from the fulcrum of the blade to the overlap of the blade) for each angle is 0.020 N · m at 30 degrees and 0.017 N · m at 50 degrees. If the opening and closing angle of the scissors is wide, the obtained reaction force will increase (Figure 2A).

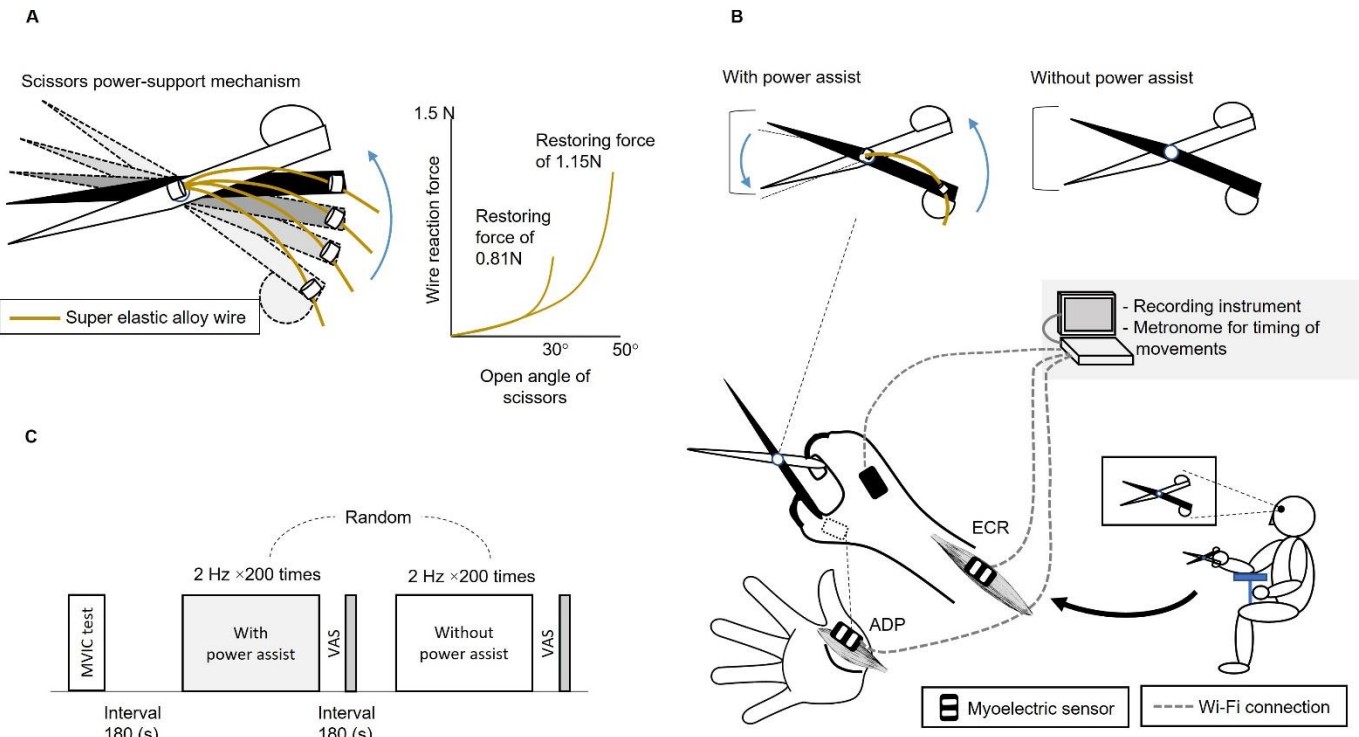

**Figure 2.** The developed mechanism of the scissors with a power-support device, and the protocol of the experiment. (**A**) Model diagram of the power-support mechanism of the scissors with power-support device. (**B**) Surface electromyogram (sEMG) data acquisition method. The myoelectric sensor was placed on the extensor carpi radialis (ECR) and adductor pollicis (ADP) muscles of the right forearm, and the myoelectric potential obtained from the sensor was analyzed. The recording computer and sensor were connected to Wi-Fi, and a metronome sound was outputted to maintain opening and closing timing. (**C**) The experimental procedure. A 180-s break was provided after performing the maximum isometric contraction task, conditions were set using a random table for the presence or absence of assistance, and the subjects asked to perform scissors opening and closing exercises.

### 2.6. Electrode Placement

The Delsys Trigno Wireless EMG system (EMGWorks, Version 4.7.3.0, Delsys Inc., Boston, MA) with a 16-bit resolution was used to detect electromyogram signals and angular velocity of thumb from the ADP and ECR during the scissor opening and closing tasks. Avanti Sensors were used as EMG electrodes, and were placed parallel to the muscle fibers with an inter-electrode distance of 10 mm. To ensure an inter-electrode impedance <20 kΩ, measured with a digital multimeter for the electrodes, the placement sites were scrubbed with cleansing gel and cleaned with 70% alcohol.

For the sEMG data, sensors were installed at two locations, ECR and ADP, on the right forearm, to verify the opening and closing operation of the scissors (Figure 2B). In this experiment, two sEMG sensors were used to measure the maximum contraction of a key pinch, the maximum isometric contraction of the ECR, and the opening and closing task of the scissors (Figure 2C).

### 2.7. EMG Data Processing

The raw sEMG data were recorded using Delsys EMGworks Acquisition software (Common Mode Rejection Ratio: CMRR > 80 dB at 60 Hz; gain of 300; band pass filtered 20–450 Hz) and were sampled at 2000 Hz. Signal analysis was conducted using Delsys EMGworks Analysis software. Three seconds after the start of the maximum isometric contraction measurements of key pinch and ECR, an intermediate 3-s interval was selected for analysis. In the scissors opening/closing task, the middle 80 s during the 100-s task was analyzed.

The digital signal of the EMG amplitude before analysis was expressed as EMG root mean square (RMS) values with a window size of 1 s. To detect the maximum RMS amplitude in the MVC, we used a window size of 1 s, moving in 100 ms increments across the MVC. In the scissors opening and closing task, the RMS value was determined and normalized using the maximum RMS detected in each muscle MVC, representing the relative %MVE. The event was set for the entire period and analyzed with a 1-s epoch.

An APDF was used to evaluate the muscle activity over the total window, with muscle activity identified in the APDF as the 10th (P0.1), 50th (P0.5), and 90th (P0.9) percentiles of the entire recording period for the normalized RMS processed signal, and summarized for each level as the mean [22]. The percentile was represented as an indicator of static muscle activity (P0.1, low level amplitude over 90% exercise time), central muscle activity (P0.5), and peak levels of muscle activity (P0.9, high levels amplitude).

### 2.8. Statistical Analysis

To verify the effect of the power-assisted scissors, an APDF analysis calculated the average percentile for each subject. The calculated values were compared using paired t-test under the conditions P0.1, P0.5, and P0.9. The difference between the presence or absence of assistance and each condition of APDF was added by two-factor, repeated-measures analysis of variance (ANOVA). The difference in pain and subjective fatigue between the ADP and ECR, depending on the assistance condition, was compared by Wilcoxon signed-rank test. All evaluations were performed using SPSS (version 26.0; IBM Corp., Armonk, NY), with the significance level set at 5%.

### 3. Results

#### 3.1. Subject Characteristics

A total of 123 rehabilitation students were sent an email requesting their participation in the study, of which 21 agreed and met the research selection conditions. One student was excluded because his score was less than 80, as determined by the EHI. Therefore, 20 students were included. The height, weight, BMI, EHI score, CTSI-JSSH score, and key pinch strength (N) of the subjects are shown in Table 1.

**Table 1.** Participant characteristics.

| Survey Items | Classifications | Participant Ratio | |
|---|---|---|---|
| | | *n* | % |
| Age, years | $18 \leq 19$ | 2 | 10 |
| | $20 \leq 29$ | 17 | 85 |
| | $30 \leq 39$ | 1 | 5 |
| Height, cm | $149\leq$ | 1 | 5 |
| | $150 \leq 159$ | 11 | 55 |
| | $160 \leq 169$ | 7 | 35 |
| | $170 \leq 179$ | 1 | 5 |
| Weight, kg | $49\leq$ | 8 | 40 |
| | $50 \leq 59$ | 11 | 55 |
| | $60>$ | 1 | 5 |
| EHI score | $80 \leq 89$ | 3 | 15 |
| | $90 \leq 100$ | 17 | 85 |
| CTSI-JSSH score | $0\leq$ | 20 | 100 |
| | | Mean | SD |
| Body mass index (kg/m$^2$) | | 20.93 | 1.36 |
| Key pinch strength (N) | | 57.33 | 17.31 |

SD, standard deviation; EHI, Edinburgh Handedness Inventory; CTSI-JSSH, The Japanese Society for Surgery of the Hand version of the Carpal Tunnel Syndrome Instrument.

*3.2. Difference Due to Assistance of Power-Support Scissors*

Table 2 shows the VAS score of pain and fatigue, and the muscle activity level of each muscle, with and without power support. The ADP showed significantly lower muscle activity with and without power support ($p = 0.004$). The ECR did not show a significant difference in muscle activity under each condition. VAS score was found to be significantly lower for fatigue and pain of the thumb and radial sides of the forearm under the assisted power condition.

**Table 2.** Difference in variables due to the assistance of power-support scissors.

| Measure | With Assist (*n* = 20) | | Without Assist (*n* = 20) | | Statistics |
|---|---|---|---|---|---|
| **VAS** | **Mean** | **SD** | **Mean** | **SD** | |
| Thumb Fatigue | 2.25 | 1.29 | 3.50 | 1.82 | [a] P = 0.008, [b] r = 0.60 |
| Thumb Pain | 0.80 | 1.06 | 1.45 | 1.54 | [a] P = 0.031, [b] r = 0.51 |
| Rad Fatigue | 4.35 | 1.79 | 5.75 | 2.07 | [a] P = 0.021, [b] r = 0.58 |
| Rad Pain | 1.95 | 1.79 | 3.20 | 1.96 | [a] P = 0.021, [b] r = 0.61 |
| **Mean %MVE (Right Side)** | **Mean** | **SD** | **Mean** | **SD** | |
| ADP | 0.14 | 0.05 | 0.15 | 0.09 | [a] P = 0.021, [b] r = 0.58 |
| ECR | 0.23 | 0.07 | 0.26 | 0.10 | [a] P = 0.124, [b] r = 0.36 |

For statistical analysis, the Wilcoxon signed-rank test was used for VAS and %MVE (ADP) after the normality test, and the paired t-test was used for %MVE (ECR). [a] Test between two groups due to difference in assistance of power-assist scissors. [b] Effect size in each test. VAS, visual analog scale; Rad, radial side forearm; MVE, maximal voluntary electrical activity; ADP, adductor pollicis muscle; ECR, extensor carpi radialis; SD, standard deviation.

An APDF showed significant differences in ADP muscles overall for P0.1, P0.5, and P0.9, indicating that ADP muscle activity was reduced by the support mechanism of the power-assisted scissors (Figure 3A). The ECR without power assistance showed the highest peak load (P0.9) (mean: 39 %MVE) in overall muscle activity (Figure 3B). Power-assisted ADP was found to be 1.9% lower at the P0.1 level, 3.3% lower at the P0.5 level, and 6.5% lower at the P0.9 level. The ECR showed no change at any activity level. The relationship between percentiles with and without assistance was verified by ANOVA. As a result, no interaction effect was observed between the assisted conditions, and the main effect of the percentile was observed (ADP: $F_{[1.15, 40.16]} = 104.43$, $p < 0.01$; ECR: $F_{[1.05, 36.79]} = 102.88$, $p < 0.01$).

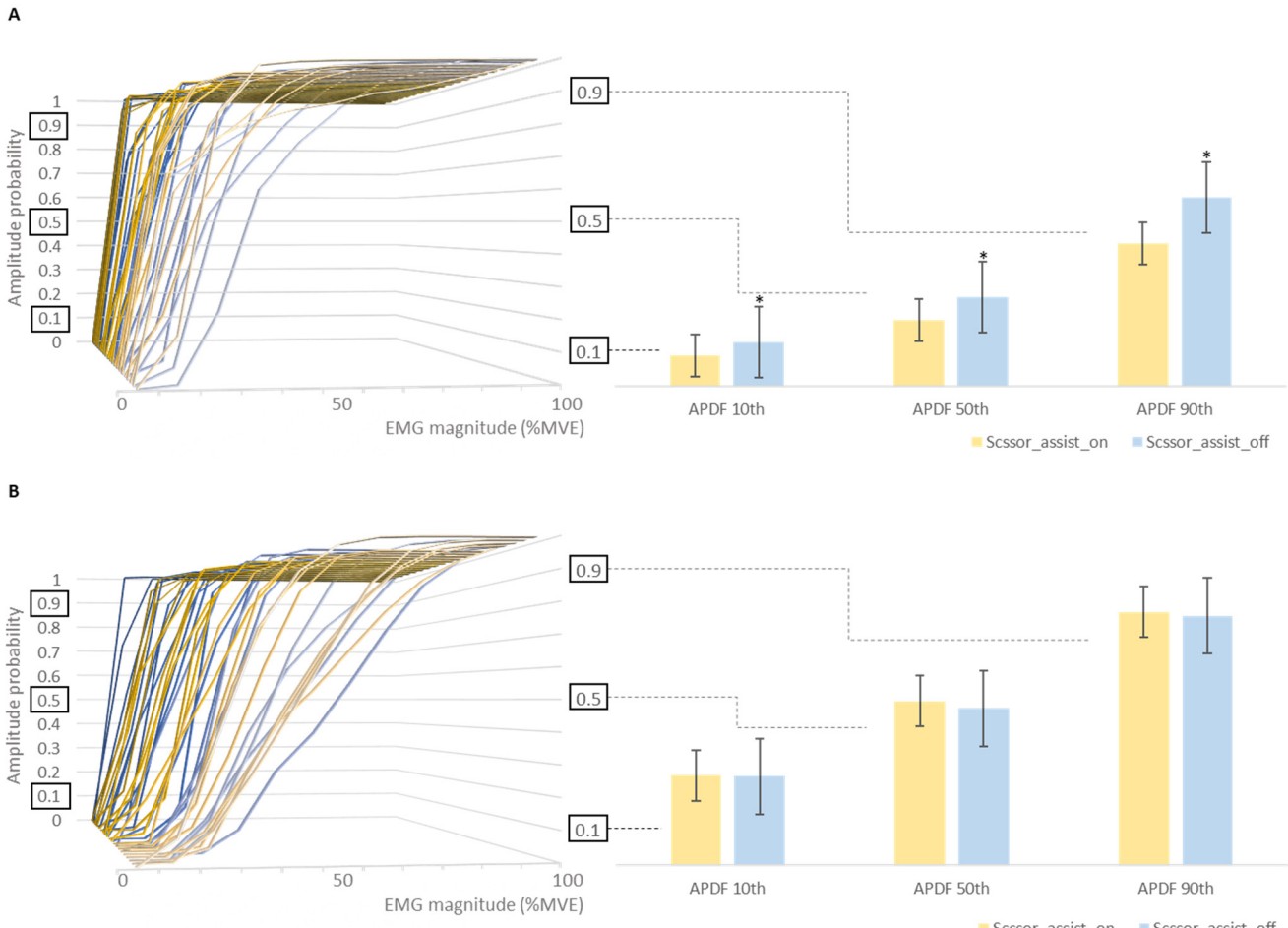

**Figure 3.** Comparison of muscle activity levels with and without power assistance, by plotting the probability of appearance of % maximal voluntary electrical activity (MVE) in the adductor pollicis muscle. (**A**) Amplitude probability distribution function (APDF) analysis plots of the adductor pollicis muscle (ADP) are shown, and muscle activity compared. (**B**) Amplitude probability distribution function analysis plots of the extensor carpi radialis (ECR) muscle are shown, and muscle activity compared. * $p < 0.05$. The polygonal line (on the left side of the A and B panels) represents the APDF value calculated from the surface electromyogram (sEMG) during scissors operation, and when the maximum pinch force was generated (y-axis, amplitude probability; x-axis, %MVE). Amplitude probability distribution function analysis shows the mean of the 10th percentile, 50th percentile, and 90th percentile of %MVE. The yellow line indicates with assistance; blue line, without assistance; and the values of P0.9 were used to rearrange and display each in ascending order. The bar graphs (to the right of **A**,**B**) compare muscle activity at each percentile by assistance condition. Error bars show the mean standard error.

## 4. Discussion

In this study, we investigated the myoelectric potential activity, subjective fatigue, and pain of the ADP and ECR when power was assisted during scissors opening and closing movements using healthy women as subjects, and compared the differences with unassisted scissors. As a result, when using scissors with a power-support device, the active muscle units of the ADP were lower in static load level (P0.1), average load level (P0.5), and peak level (P0.9) than when using scissors without power assistance. There was no statistically significant difference in ECR muscle activity. Subjective fatigue and pain when using scissors with power support was lower than when using scissors without. These results support the hypothesis that scissors with a power-support device can reduce muscle fatigue and hand pain of the ADP during exertion. These findings may help hairdressers prevent the development of occupational diseases. However, as the subjects

within the study were not professionals, it remains necessary to verify whether a power-assist device can prevent hand disorders, such as CTS, for hairdressers without interfering with cosmetology techniques.

The results of this study showed that the average load level (P0.5) %MVE in the ADP while manipulating the scissors was 9.4 %MVE with assistance and 12.7 %MVE without assistance, and the muscle output was reduced to 10 %MVE or less by power assistance. The support mechanism of the power-support scissors is powered by the restoring force generated from the strain of the stretched wire, and assists the power of the lower blade of the scissors. In the preliminary experiment, the force to bring the lower blade closer to the upper blade was confirmed (0.81 N). Jeong et al. showed that assisting finger movement with wire tension can reduce sEMG muscle activity [23]. From these, and our, findings, it can be surmised that the restorative force of the steel wire used in the power-support device of the scissors decreased the burden on the muscles of the thumb. In addition, Xiong and Muraki measured electromyographic data while operating a touch screen with the thumb, and found that the muscle activity of the ADP muscle increased in the direction of adduction–abduction [24]. The opening and closing movement of the scissors in this study was a combined movement of the thumb, with the wrist fixed in dorsiflexion of about 30 degrees, accompanied by flexion-palmar abduction-palmar adduction. This movement is repeated adductor pollicis abduction, and it is speculated that scissors with a power-support device could effectively assist the muscle output of the ADP.

Hairdressing involves repetitive tasks, and it has been reported that women's haircuts take an average of 51.4 min [3]. It has also been reported that long-term repetitive work causes fatigue and pain in the entire upper limb [2]. However, fatigue and pain decreased with our power-assist device. Jonsson pointed out that, in vocational training tasks when working at an average load level (P0.5) of 1 h or more, muscle load becomes excessive when the %MVE value exceeds 10% [25]. The %MVE at the average load level (P0.5) in this study was 9.4 %MVE (with assistance) and 12.7 %MVE (without assistance), and the muscle output was reduced to 10 %MVE or less by power assistance. Chen et al. reported that the static load level (P0.1) was 2–6 %MVE when measuring the muscle activity of the extensor/flexor muscles of the forearm during haircutting movements of a female hairdresser [3]. The %MVE of the static load level in this study was similar (6%), but was attenuated by approximately 2% with power assistance. These results also indicate that the power-support device compensates for the muscle burden. Fatigue associated with muscle activity results in a temporary decrease in skeletal muscle strength and capacity [26], with impaired activation of the motor neurons that drive the muscle fibers [27]. Therefore, it is inferred that the attenuation of the load due to the assistance maintained the skeletal muscle resources of the ADP, and fatigue was less likely to occur. In a report on cosmetologist work and risk, Hanvold et al. found that sustained muscle activity correlates with pain [28], and pain due to repetitive movements increases inflammatory markers due to muscle damage [29]. Controlling the muscles that participate in movement is effective in the prevention of muscle damage [30]. Therefore, it can be inferred that pain in the hands may be reduced as a result of decreasing the muscle burden with scissors equipped with a power-support device.

This study had several limitations. First, the participants in this study were female university students, and not cosmetologists. The results of this study cannot be applied to hairdressers, as it has been reported that the average working hours of hairdressers is 8 to 12 h per day [3,31], and the experimental time in this study was 100 s. Compared to the subjects of this study, hairdressers routinely work with scissors for long periods, so there is a difference in endurance and skill levels. Therefore, in future studies, hairdressers should be used as subjects, and an analysis comparing number of years of employment and experimental time should be performed. Second, muscles and joints other than the ADP and ECR were not included in the analysis. The angle of the wrist and shoulder joints affects the load on the fingers when operating scissors [32], and tension from the neck to the back tends to increase as a result of working in the same position [2]. In this study,

to determine the effect of power support, the measurement conditions were minimized, and wrist joint dorsiflexion and shoulder joint position were kept constant. To maximize the effect of power-support scissors and prevent occupational diseases of hairdressers, it is necessary to create an optimal model of the working limb position, including the posture-control muscles, and estimate the burden on the fingers. Third, the force of the power-assist scissors was fixed. A mechanism that can adjust the force according to the muscle strength of the fingers and the size of the hand should be developed. Fourth, in this study, we performed a simple opening and closing operation with barber scissors of a general shape, and did not verify whether power support can be obtained for each type of scissors, opening and closing angle, and procedure. Hairdressers use different tools and techniques such as using thinning scissors and cutting with a slight movement of the cutting edge. In order to extend the effect of the power support scissors we have developed, new verification with tools and techniques should be done. Fifth, the analysis of hair cutting pace was not included to optimize the effect of power support. The power support mechanism uses the restoring force of steel wire, and the effect obtained may differ depending on the cutting pace. The frequency band of the opening and closing operation of the scissors should be expanded and verified.

## 5. Conclusions

The results of this study suggest that the action of scissors with a power-support device compensates for the muscle load on the thenar eminence. This reduction in muscle strain was shown to reduce subjective fatigue and pain associated with operating scissors. Scissors with a power-support device may assist hairdressers in the future and reduce the burden on the musculoskeletal system.

**Author Contributions:** Conceptualization and design, K.K., K.S., A.O., K.T. and T.H.; acquisition of data, K.K. and T.H.; analysis and interpretation of data, K.K., Y.K. and K.S.; drafting of the manuscript, K.K.; critical revision, T.H. All authors have read and agreed to the published version of the manuscript.

**Funding:** This research was funded by JSPS KAKENHI, grant number JP19K11419.

**Institutional Review Board Statement:** The study design was conducted according to the guidelines of the Declaration of Helsinki, and approved by the Ethics Committee of Saitama Prefectural University (#30069, 2018).

**Informed Consent Statement:** Informed consent was obtained from all subjects involved in the study.

**Data Availability Statement:** The datasets generated during and/or analyzed during the current study are available from the corresponding author on reasonable request.

**Acknowledgments:** We would like to thank all the students of Saitama Prefectural University who participated in the research, and the staff of Okino Kogyo Co., Ltd. (Akihisa Okino, Kazuhisa Takeda) for their efforts in device development.

**Conflicts of Interest:** The authors declare no conflict of interest.

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
