# Peer review of "Usefulness of Scissors with a Power-Support Mechanism to Assist Thumb Movement: An Observational Study"

_applsci, doi:10.3390/app11167756_

Round 1
Reviewer 1 Report
General comments
This manuscript aims at determining whether scissors with a power-support device can improve the repetitive movement of the thumb to reduce the amount of myoelectric potential activity and examining the effect of power-supported scissors on muscle fatigue and pain. Despite some specific and minor issues detailed below, authors manage to fulfil sufficiently their aims.
Specific comments
(line 27 and elsewhere throughout MS) “beautician” or “hairdresser”?
(l112) “empty cut” condition was chosen. Could you estimate “empty cut” and “ecological cut” conditions torques?
(l113) 2.0 Hz pace was chosen. Could you estimate “ecological” cutting pace?
Did you check for blade deterioration over testing?
Minor comments
(lines 16-17) Please, re-phrase;
(l37 and elsewhere throughout MS) please, do not start sentences with acronyms;
(l41÷5 and 147÷51) please, split;
(l290) … Muraki measured… [23].
Author Response
Reply,
Reviewer #1: We wish to express our appreciation to the reviewers for their insightful comments on our manuscript. The comments have helped us significantly improve the manuscript.
Please see the attachment.

Reviewer 2 Report
General Comments
The authors aimed to examine whether scissors with a power support device can reduce the burden of the finger muscles and this research would add novel information to the research area.
Specific Comments
Abstract
Page 1, line 12: ‘is’ should be ‘was’
Page 14: delete right-handed
Introduction
Page 1, line 40, 41: I am not sure what the ‘median position’ means. Is this neutral position?
Page 2, line 49-50: this movement is mainly performed by the thenar eminence and flexor pollicis longus -The authors should include references for these studies to support this statement
Page 2, line 58: What does locomotor means? It would be beneficial to define the term ‘locomotor’.
Page 2, line 59: “Closing” or “cutting”
Page 2, line 67: Please describe more why the author used APDF analysis?
Materials and Methods
Page 2, line79-80: Why the author used the muscle activity of ADF and ECR as indicators? Not use myoelectric activity of thenar eminence and flexor pollicis longus as describing in Introduction section?
Page 3, line 104-105: The handle ring was adjusted according to any specific criteria? Please verify.
Page 3, line 114-115: What is criteria of the finger muscle fatigue? How did the author know that finger muscle fatigue after performing 100 consecutive open/close movement?
Page 3, line 135: I don't think VAS can measure fatigue of ADF and ECR precisely.
Page 4, line 137: Lots of noise were found in the data of thumb angular velocity.
Did the author describe how to collect angular thumb velocity data in Materials and Methods?
Page 5 and Page 7: The caption for figure 2 and figure 3 are too long respectively and hard to follow. Please consider alternative strategies.
Results
Page 6, line 221: Please delete the column of P-value in Table 1.
Page 6, line 223: Total the same data between two conditions? But p<0.05?
What is "r" stand for?
Discussion
Page 8, line 272-273: This sentence is awkward, please correct.
Page 8, line 287-289: Please rewrite this sentence.

Author Response
Reply,
Reviewer #2: We wish to express our deepest appreciation to the reviewers for their insightful comments on our paper. We feel the comments have helped us significantly improve the paper. In particular, we wish to acknowledge these highly valuable comments on outcome measures and analyses.
Please see the attachment.

Round 2
Reviewer 2 Report
The authors have done a good job of addressing my previous concerns. However, I still have some certain aspect that the authors need to address.

Author Response
We greatly appreciate all comments and feedback from the reviewer #2.
Reviewer #2: We wish to express our appreciation to the reviewers for their insightful comments on our paper. The comments helped improve our treatise correctly.
Comment 1. Page 1, line 41: ‘median’ should be neutral
Response to comment 1. We have changed “median” to “neutral”. (P1, line 41)
Comment 2. Page2, line 59-60. ….the load on the forearm and fingers due to the repetitive exercise was not what?,…..This sentence is awkward, please correct.
Response to comment 2. We have changed text “Although the burden was reduced, the load on the forearm and fingers due to the repetitive exercise was not, and the problem remained.” to “Although the pain in the wrist joint was alleviated, the load on the forearm and fingers due to repetitive exercise was not verified, and the problem remained.” (P2, lines 59–60)
Comment 3. Page 3, Line 143 ‘MVC’ should be ‘MVIC’
Response to comment 3. We have changed “MVC” to “MVIC” (P3, line 143).
Comment 4. Page 6, line 245 Please correct the r denotes in Thumb Fatigue and Thumb Pain measurement.
Response to comment 4. We fixed correctly "r" of Thumb Fatigue and Thumb Pain. (P6, line 245, Table 2)